# Virtual Reality as a Promising Tool Supporting Oncological Treatment in Breast Cancer

**DOI:** 10.3390/ijerph18168768

**Published:** 2021-08-19

**Authors:** Ewa Zasadzka, Anna Pieczyńska, Tomasz Trzmiel, Katarzyna Hojan

**Affiliations:** 1Department of Occupational Therapy, Poznan University of Medical Sciences, 60-781 Poznan, Poland; ezasad@ump.edu.pl (E.Z.); pieczynska.ann@gmail.com (A.P.); ttrzmiel@ump.edu.pl (T.T.); 2Department of Rehabilitation, Greater Poland Cancer Centre, 61-866 Poznan, Poland

**Keywords:** oncology, virtual reality, physical therapy, rehabilitation, supportive care, cancer care

## Abstract

Breast cancer (BC) treatment is associated with many physical and psychological symptoms. Psychological distress or physical dysfunction are one of the most common side effects of oncological treatment. Functional dysfunction and pain-related evasion of movement may increase disability in BC. Virtual reality (VR) can offer BC women a safe environment within which to carry out various rehabilitation interventions to patient support during medical procedures. The aim of this systematic review was to conduct an overview of the clinical studies that used VR therapy in BC. The review was conducted in accordance with the Preferred Reporting Items for Systematic Reviews and Meta-Analyses (PRISMA) guidelines method: the initial search identified a total of 144 records, and 11 articles met the review criteria and were selected for the analysis. The results showed that VR seems to be a promising tool supporting oncological treatment in BC patients. VR can have a positive effect on mental and physical functions, such as relieving anxiety during oncotherapy, diminution pain syndrome, and increasing the range of motion and performance in daily activities.

## 1. Introduction

Breast cancer (BC) is the most commonly occurring cancer in women over the world [1,2]. It severely affects both physical and mental health. Psychological stress is the most common consequence of BC diagnosis [3] and treatment, and a frequent symptom of anxiety and depression [4]. Oncological treatment of BC involves choosing the right intervention based on a number of prognostic and predictive factors. The most important ones are the stage of clinical advancement, the presence of metastases, the histological type, the biological subtype, or the results of molecular tests. Standard cancer treatments include surgical treatment, chemotherapy, and radiation. These may lead to further life quality deterioration due to side effects they produce [4]. Additionally, pain may be accompanied by both physical fitness and mental health deterioration, regardless of disease stage [5].

Effective supportive therapies have the potential to reduce the negative effects of treatment and the cancer recurrence risk and mortality rate [6]. However, significant side effects and life quality deterioration are affected in these oncological treatments [7]. Currently, BC surgery may involve physical and pain symptoms of the posture and upper extremities [8,9]. Functional dysfunction and pain-related evasion of movement may increase the disability level in BC. Additionally, axillary radiotherapy is associated with increased risk of posture and upper limb impairment such as lymphedema, shoulder, and arm mobility dysfunction, and soft tissue thickening [8]. Other symptoms such as fatigue, pain, a decrease of muscle strength or range of motion (ROM), developed in the postoperative time result in daily living activity limitations [10]. Physical exercises, manual mobilisation and stretching involving a combination of physiotherapy technics are effective in postsurgical pain management and restoring limb functionality [11,12,13]. Next, the lymphedema of the upper limb is one of the most frequent morbidity causes after mastectomies with axillary lymph node dissection and radiation [8,11,14]. This condition involves abnormal accumulation of fluids and proteins in the intercellular space, chronic inflammation, and oedema [14] which consequently causes upper limb functional impairment. Additionally, limitations in ROM following breast cancer treatment may be due to scar tissue formation at the incisional site, post-radiation fibrosis, protective forward shoulder posturing, post-mastectomy pain syndrome, and/or restriction of use upper extremity [8,14]. Limited shoulder ROM may restrict activities of daily living, e.g., tilting the head back while washing or combing hair, or arm movements involving shoulder elevation during, e.g., reaching for the item high on the shelf [15]. Traditional physiotherapy, the use of kinesiotherapy, and massage-using modalities concentrate on improving upper limb mobility, while task-oriented programs focus on BC patients’ adaptability to daily life situations. They aim at shortening adaptation processes and gaining problem-solving processes of daily tasks by taming the dread of movement. 

Task-oriented programs seem to be superior to exercises involving repetitive multiple movement patterns. Task-oriented training programs also improve patients’ quality of life and functionality [16]. There is a rapid increase in the use of new technology in rehabilitation as tools that motivate patients to participate in usual care and standard programs. Virtual reality (VR) therapy seems to be as effective as conventional therapy in improving upper extremity function and basic activities of daily living [17]. These interventions are effective as patients focus on pleasant or interesting stimuli instead of focusing on unpleasant symptoms which are connected with the emotional sphere [18]. Supportive techniques using humour, relaxation, music, imagery, and VR are classified as distraction interventions, where symptoms such as pain, anxiety, nausea, fatigue, and stress may be relieved [19]. Distraction can also alleviate psychological symptoms [20].

### Virtual Reality

Recent years have seen a real change in the world due to advances in rehabilitation technology. A great variety of user-friendly pieces of equipment with multiple settings or/and functions has been developed. The use of modern technology in the health field is a source of new knowledge and offers significant advantages.

In 1989, Jaron Lanier first used the term virtual reality (VR). Then, the definitions of VR referred to a specific technological system, which typically includes a computer capable of real-time animation controlled by wired gloves or other controllers and a position tracker, as well as a head-mounted display for visual yield [21]. Many different definitions of VR are available in the literature. One of a recent [22], defines VR as ‘an artificial environment that is experienced through sensory stimuli such as images and sounds provided by a computer and in which our actions partially determine what happens in the environment’.

VR systems are divided into two categories: immersive and non-immersive [23]. Immersive VR is defined by filled immersion, which can affect users’ attention. A whole-immersive experience is achieved through a head-mounted exhibition that obscures users’ view of the real world and presents patients with a computer-generated view instead. The head-mounted display and headphones exclude access to light and sound from the outside. Thus, patients can be personally isolated from the hospital–medical milieu. This is what helps patients to focus on enjoyable stimuli and reduce unfavourable emotions [24]. Jennett et al. [25] defined immersion as a ‘lack of awareness of time, a loss of it’. A key driver of distraction in VR is its capability to simultaneously involve various senses, delivering synthetic stimuli such as visual images, spatial sound, or tactile and olfactory feedback [19]. 

The non-immersive VR is characterised by a computer screen, where the user is joined to the virtual world but can still communicate with the external environment. This technology offers patients a safe environment, where it is possible to carry out a variety of interventions including lifestyle changes [26], rehabilitation at home, or providing support to hospitalised patients undergoing various medical treatments. VR is emerging as a promising device to support cancer patients and monitor neurophysiological changes and medical feedback during interventions [23].

The aim of this systematic review was to conduct an overview of the clinical studies where researchers used VR intervention in BC patients as a tool in cancer rehabilitation of this patients’ group.

## 2. Materials and Methods

The guidelines of Preferred Reporting Items for Systematic Reviews and Meta-Analyses (PRISMA) [27] were used to conduct the review. Ethics Committee approval for this type of retrospective study was not required. This review was registered in Research Registry (reviewregistry1152).

### 2.1. Search Strategy

The identified keywords and Medical Subject Headings (MeSH) were combined using the following combinations of terms with the Boolean operator ‘AND’: ‘virtual reality’, ‘VR’ AND ‘breast cancer’ was used to find relevant publications. This systematic review was conducted by querying PubMed, Web of Science, PEDro, and Cochrane Library databases were searched from the establishment of the database to 1 March 2021.

A search for additional articles was also carried out by browsing through the reference lists. The authors (EZ and AP) conducted an independent search. 

### 2.2. Outcome Measure

#### 2.2.1. Primary Outcome

The primary outcome was the analysis of using VR systems on the physical functions of BC patients.

#### 2.2.2. Secondary Outcome Measure

The secondary outcome measure was the analysis of VR treatment on the mental sphere and the pain level in BC patients.

### 2.3. Data Collection and Analysis

An independent review and analysis of the articles were undertaken by the authors (E.Z. and A.P.), who removed duplicates and then checked for compliance with the inclusion and exclusion criteria. The following data were extracted from the articles: first author, year of publication, study population characteristics, study design, inclusion/exclusion criteria, intervention characteristics, assessment of the outcome, and results.

### 2.4. Inclusion and Exclusion Criteria for the Articles

The inclusion criteria for the reports were as follows: published in English in a journal with a review process; original research study with a control group or/and presentation of results of comparative pre- and post-therapy involving VR in BC; clearly defined inclusion and exclusion criteria for the study groups.

The following articles were excluded: studies on populations including other patients than BC; animal studies; studies examining the effect of robotic intervention; studies lacking the approval of local ethics committee or with incomplete outcome data; studies of undetermined type, and pilot studies or conference proceedings.

### 2.5. Quality Assessment

The quality of the included studies was assessed using the Quality Assessment Tool for Quantitative Studies (QATQS) [28] by the authors (E.Z. and A.P.). Possible conflicts were discussed and resolved by the other author (K.H.). The following sections were assessed: selection bias; study design; confounders; blinding; data collection methods, withdrawals and dropouts; intervention integrity; and analysis, classifying them as ‘weak’, ‘moderate’ or ‘strong’, according to a reviewer’s key. The rating of one section as ‘weak’ results in the evaluation of the entire study as ‘moderate’. If more than one rating is ‘weak’, the survey will automatically determine ‘weak’. Lack of the ‘weak’ rating in individual sections allows the evaluation of the entire study as ‘strong’.

## 3. Results

### 3.1. Evaluation of the Study

In total, 137 records were identified from searches in all databases, and 9 additional records were identified after the reference list search; a total of 11 entries [29,30,31,32,33,34,35,36,37,38,39] were included in the study. The PRISMA flowchart of the search process is presented in Figure 1. 

### 3.2. Methodological Quality

The methodological quality of all included studies was presented in Table 1. Seven studies were considered as ‘strong’ [29,30,31,33,34,35,39], and four were considered ‘weak’ [32,36,37,38]. The highest-rated section was data collection, and the worst sections were blinding and selection bias.

Figure 2 presents a detailed assessment of individual sections expressed as a percentage. 

### 3.3. Characteristics of Study Participants

The total number of participants in the analysed studies was 619. In four studies [29,30,31,34], participants were assigned to a control group (CG) and an intervention group (IG). Three studies did not include CG [32,33,35], and the remaining four [36,37,38,39] used a cross-study type design where the same patients constituted CG and IG. All women were diagnosed with BC. Participants’ characteristics of the included studies are presented in Table 2.

Three of the studies also included patients with BC and other types of cancers such as gynaecological cancer [33], colon, or lung cancer [38,39]. In four studies [29,31,32,35], VR-based therapy was used as an adjunctive treatment after surgical removal of breast tumours, and in six [30,33,36,37,38,39] during chemotherapy.

### 3.4. Characteristics of VR Interventions, Outcome Measure, and Study Results

Interventions with the use of VR in the studies analysed can be divided into those that affected the physical functions of patients [29,31,32,35], the mental sphere [30,33,34,36,37,38,39], and the pain level [14,32,34].

Atef et al. [29] compared VR therapy (using video games of tennis, triceps extension, and rhythmic boxing) and the PNF method in the treatment of lymphedema and improving upper limb functionality. Their results indicate a statistically significant improvement in both study variables after the VR intervention. Feyzioğlu et al. [31] also used sports games (darts, bowling, boxing, beach volleyball, table tennis) for therapy to compare their effectiveness with standard physiotherapy. The authors examined shoulder range of motion, arm and handgrip strength, upper extremity functionality, fear of movement, and pain intensity. House et al. [32] conducted motor training in VR, but also emotive and cognitive training. They used the BrightArm Duo Rehabilitation System consisting of a robotic rehabilitation table, computerised forearm supports, a display, a laptop computer for the therapist, a remote clinical server, and a library of custom integrative rehabilitation games. Plejko et al. [35] used VR for exercises in patients after mastectomy, assessing their influence on static and dynamic postural control. Non-immersive VR was used in the interventions aiming at improving motor functions in patients. 

The remaining articles analysed used immersive VR. In five studies [30,36,37,38,39], VR technology was used in patients during chemotherapy by intravenous infusions. Chirico et al. [30] compared the effects of VR and music therapy on the anxiety level and mood states. Patients had a controller to interact with the virtual environment, which consisted of relaxing landscapes. For example, they were exploring walking through a forest, an island, observing different animals, swimming in the sea, and climbing a mountain. Schneider et al. [36,37,38,39], on the other hand, in their four studies offered patients VR sessions consisting of diving in the deep sea, walking over an art museum, or solving a mystery, while testing for levels of anxiety, stress, and fatigue associated with chemotherapy.

McGarvey et al. [33] investigated whether VR technology could reduce the level of stress associated with hair loss in women after chemotherapy. Through the use of VR, patients were able to see themselves with a bald head, various wigs, and hair styles.

In three publications [31,32,34], the authors assessed the influence of VR therapy on patients’ pain levels using the VAS [31,34] or NRS [32] scale. In each case, VR therapy was effective in relieving pain. Bani Mohammad et al. [34] showed that the combination of VR therapy with morphine administration provides a better analgesic effect than pharmacotherapy alone.

The results of all the studies analysed showed the effectiveness of VR-based therapy in reducing lymphoedema, improving dynamic balance and upper extremity functionality, reducing the level of stress, anxiety, and fatigue associated with chemotherapy, reducing stress levels related to hair loss after chemotherapy, and alleviating pain. Table 3 presents the type of interventions, characteristics of measures, and results of the included studies.

## 4. Discussion

This systematic review presented an overview of the clinical studies in which researchers used VR intervention in BC patients. Despite the interpretability of the findings, this review is limited by the inclusion of a small number of trials given the novelty of this approach, generally small sample sizes, and diverse design of trials (including data from single-arm studies). Seven studies included in the present systematic review had a good level of evidence (low risk of bias). Additionally, some of the revised studies did not include information about the characteristics of VR rehabilitation protocols, reducing the possibility of replication by next studies.

Over the past two decades, new technologies such as VR, in which users to be immersed into a three-dimensional world in the computer, have found various applications in health care [19], offering both immersive and non-immersive experiences. 

Numerous research studies have shown that a sitting lifestyle is now one of the considerable health problems associated with many diseases, including hypertension, cardiac diseases, metabolic disorders, cancer, and mental illness. Although the WHO recommends engaging in physical activity, a very large number of patients lead a sedentary lifestyle despite knowing about the benefits of regular exercise on general health. One of the factors contributing to this occurrence is the motivation deficiency in people as a barrier for intending to start changes in health habits. New alternatives to exercise such as VR can help patients lead healthier lifestyles [40].

The semi-immersive system is also used in the rehabilitation of various patient groups. The studies by De Luca et al. [41,42] show a positive effect of the exercises using semi-immersive VR in patients after stroke and after brain injury. The main advantage of VR is creating a motivating environment, with interactive and multisensory stimulation [43,44]. Specifically, this system is made up of computerised software, several tagless sensors, a video camera, and a projector connected to a screen. This installation allows an individual to perform different exercises; the participant exercises in a virtual context to inspire many cognitive spheres through a screen interface that responds to the patient’s movements with audio–visual feedback. In practice, the patient’s movements are monitored by an imaging camera that reinforces the information to the PC program, allowing the screenplay to be changed. In addition, this enables a greater consciousness of movements and performance and allows for sensory involvement with effects on rehabilitation outcomes [45].

As previous research shows [23], immersive VR is particularly recommended as a distraction instrument in relieving pain, stress, and other side emotional effects during different medical procedures, including chemotherapy in cancer patients. Emotional instability may be one of the factors that may increase the length of stay at the hospital during the treatment procedure or increase the quantity of sedation required during a painful procedure. Moreover, a stressed patient does not cope with treatment very well and may have difficulties in cooperation with the health service, thus making it difficult to carry out the procedure [46]. The authors of this review [23] suggested that immersive VR supports promise as an effective abstraction intervention in the treatment of pain and anxiety amongst patients with BC. Exertions that may reduce treatment-related stress or anxiety should intensify a person’s capability to muddle through the disease, not merely by eliminating a stressor but possibly also helping to create a feeling of being in control of the disease. Another factor that can cause various problems during cancer treatment is hospitalisation. It can be considered a stressful state itself because it results from a shift in health and may also entail stressful conditions, such as a lack of intimacy and feelings of uncertainly. Espinoza et al. [47] have demonstrated the effectiveness of VR as a therapeutic tool to deal with a variety of problems and patient needs. Successful disease management can lead to improved adherence to treatment systems due to improvement of patients’ survival time and their daily quality of life [36].

In the presented research, non-immersive VR was shown as more applicable in the rehabilitation treatment of BC patients. A study conducted by Feyziogli et al. [31] shown VR exercises using the Xbox Kinect™ console as more efficient than standard physical therapy in treating upper limb dysfunction after BC surgical treatment. Xbox Kinect™ video games are able to provide fun, inexpensive, and motivating exercise programs. Those authors [31] suggested that Kinect-based VR rehabilitation programs should be added to standard physical therapy or recommended as a substitute for traditional physical exercises for BC patients especially with severe levels of anxiety or pain about movement following surgery [31,32,34]. A study by Atef et al. [29] indicated VR as a beneficial tool in reducing lymphedema after mastectomy, which may be used as an exercise-based technique in BC patients as it motivates and provides visual feedback to them.

Exercises using non-immersive VR, which may stimulate the brain and effectuate motor and cognitive responses at the same time, require activation of the cortical and subcortical circuits [48]. Thanks to technological progress, new games are created, especially interactive games, which have the ability to improve patients’ balance [49], physical fitness, and speed processing, and executive functions [50]. Changes in motor functions [51] may be related to the reorganisation of the cerebral cortex [52]. This was shown by previous studies on stroke patients who were treated with this technology [52].

Cancer rehabilitation is a relevant part of cancer patient care, which is becoming more and more essential with the growing quantity of cancer survivors, and highly documented rates of disability [53]. For example, the COVID-19 pandemic has required a modification from personal contact during rehabilitation to virtual care using telephone visits or the application of new technologies including VR [54]. However, this alteration has largely been made without sufficient evidence of best medical practice. Rehabilitation using VR may improve access for geographically scattered patients [55]. VR has emerged as an effective new approach to rehab treatment in various health areas [36], in the promotion of emotional well-being in hospitalised patients [56], diagnostics [57], surgical training [58], as well as in mental health treatment.

The limitations found in the revised studies indicate that despite the increased use of VR technology in cancer rehabilitation, it is not possible to draw strong conclusions about VR-based rehabilitation for BC patients because of the overall lack of methodological quality and statistical power observed in the current literature. Future research should avoid methodological limitations and use and report adequate statistical results so as to identify the effects of VR training and assure robustness for proper quantitative data analysis.

## 5. Conclusions

The review showed that VR seems to be one of the promising tools supporting oncological treatment in BC patients. VR can have a positive effect on mental and physical functions, such as relieving anxiety during oncotherapy, reducing pain syndrome, lymphoedema, and improving the range of motion and performance in daily activities. Rehabilitation with the use of this equipment may be supporting in cancer rehabilitation and probably suggested for this BC group who have a problem reaching routine rehabilitation services (e.g., due to lack of transport). Additionally, it is advisable to undertake randomised, controlled trials in a large group of BC patients with the use of VR therapy during and after oncological treatment. 

## Figures and Tables

**Figure 1 ijerph-18-08768-f001:**
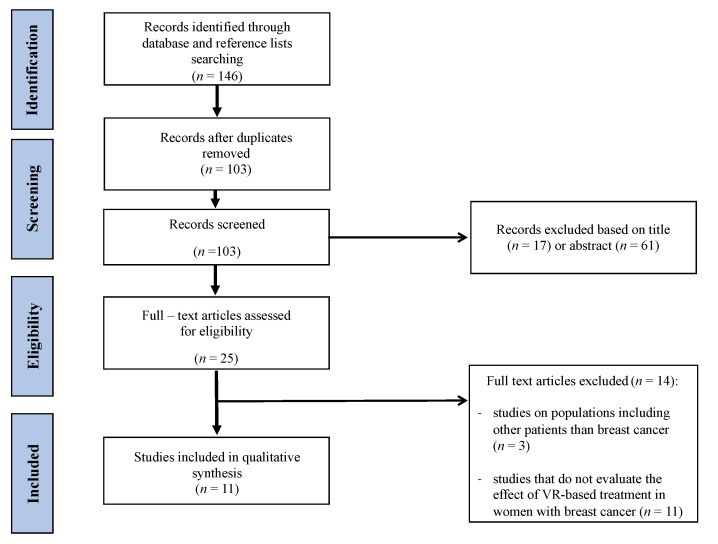
PRISMA flowchart of the search strategy process.

**Figure 2 ijerph-18-08768-f002:**
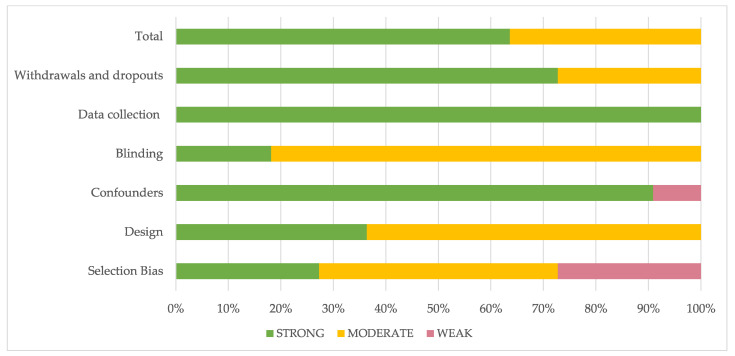
Evaluation of the individual items within the quality sections presented as percentages across all included Scheme.

**Table 1 ijerph-18-08768-t001:** The methodological quality of each study using QATQS.

Study	Selection Bias	Design	Confounders	Blinding	Data Collection	Withdrawals and Dropouts	Total
Atef 2020 [29]	1	1	1	2	1	1	1
Chirico 2019 [30]	1	1	1	2	1	1	1
Feyzioğlu 2020 [31]	2	1	1	1	1	1	1
House 2016 [32]	1	2	3	1	1	2	2
McGarvey 2009 [33]	2	2	1	2	1	1	1
Mohammad 2018 [34]	2	1	1	2	1	1	1
Piejko 2020 [35]	2	2	1	2	1	2	1
Schneider 2003 [36]	3	2	1	2	1	2	2
Schneider 2004 [37]	3	2	1	2	1	1	2
Schneider 2007 [38]	3	2	1	2	1	1	2
Schneider 2011 [39]	2	2	1	2	1	1	1

1—strong; 2—moderate; 3—weak.

**Table 2 ijerph-18-08768-t002:** Characteristics of the participants of the included studies.

First Author and Year [No. Ref.]	Number of Participants [N] and Design of Study	Age [Years]	Inclusion Criteria	Exclusion Criteria
Atef 2020 [29]	*N* = 30 (Quasi-randomised study of 2 groups: 15 women had VR-based therapy, and 15 had PNF therapy)	range: 40–65	BC women with unilateral lymphoedema after mastectomy, min. 6 months after surgery, min. 5% difference in limb volume, not undergoing physiotherapy for at least 3 months.	musculoskeletal, neurological, and visual disorders, uncontrolled cardiovascular or pulmonary diseases that are not controlled, psychiatric illness, bilateral lymphedema, elephantiasis, current metastases, continuing RT and venous thrombosis
Chirico 2019 [30]	*N* = 94 (Non-randomised study: 3 groups: 30 patients were included in the VR IG, 30 patients in MT IG, and 34 patients constituted the CG with standard care)	mean (SD): VR: 55.18 (5.7) MT: 55.7 (5.26) CG: 56.2(6.79)	age 18–70 years, CHT as a treatment for BC (Epirubicin, Cyclophosphamide)	epilepsy, drug and/or alcohol addictions, metastasis, wearing glasses, or having ports
Feyzioğlu 2020 [31]	*N* = 40 (Randomised, clinical study: 2 groups: 20 patients were included in the Kinect-based rehabilitation group (KBRG), and 20 in the standardised physical therapy group (SPTG);	mean (SD): KBRG: 50.84 (8.53) SPTG: 51.00 (7.06)	women aged 30–60 years, in the second week after surgery with axillary dissection, without vision, hearing, and speech impairment	previous BC surgery, cancer focus, reduced range of motion of upper limb before surgery, pace-maker, infection, open wounds, or wound drains; and mental disorders or cooperation issues
House 2016 [32]	*N* = 6 (Pilot and single-arm study)	range: 22–78 mean (SD): 57.8 (20.4)	age ≥ 22, regular intake of painkillers, mild to moderate depression, upper limb impairments	immobilised upper limb, visual or hearing impairments, severe cognitive problems, violent behaviour, metastases to the upper limb bones
McGarvey 2009 [33]	*N* = 45 (Simple size, evaluation study: patients were assigned to the HAAIR–IG or a control group (SCG)).	mean (SD): IG: 51.72 (10.55) SCG: 50.85 (10.31)	women with BC or other cancers treated with CHT that has been associated with alopecia	age < 17 or >75, severe physical or mental disability, previous episode related to cancer and alopecia
Bani Mohammad 2018 [34]	*N* = 80 (Randomised, controlled trial: 40 in IG and 40 in CG)	range: 30–70	women aged 18–70 years with BC, current history of chronic pain treated with morphine or other painkillers, writing and reading, no epilepsy and brain metastases, no motion sickness, no significant cognitive, visual, and hearing impairment.	-
Piejko 2020 [35]	*N* = 46 (Pilot, clinical, non-controlled study)	range: 36–63 mean (SD): 51.67(6.62)	women age > 18, after mastectomy for BC (grade I-III clinical advancement) and adjuvant treatment, completed RT and CHT a min. of 8 wks. before the start of the study, consent to participate in the study.	Barthel scale < 65 points, lymphoedema, body imbalance due to other diseases, inability to cooperate
Schneider 2003 [36]	*N* = 16 (IG and CG—crossover design. Participants received the VR treatment during either their 1st CHT treatment (group A) or their 2nd CHT treatment (group B). During the alternate CHT treatment (CG) subjects received standard care.	range: 50–77	age ≥ 50, BC, no history of other cancer, at least two matched cycles of IV CHT, reading and writing in English, no metastasis and primary brain disease, without a history of seizures, no history of motion sickness, Mini-Mental ≥ 24	-
Schneider 2004 [37]	*N* = 20 (Crossover design. Participants received the VR treatment during either their 1st CHT treatment (group A) or their 2nd CHT treatment (group B). During the alternate CHT treatment (CG) subjects received standard care)	range: 27–55	women age 18–55 with BC, no history of other cancer, require at least two matched cycles of IV CHT, reading and writing in English, no metastasis and primary brain disease, without a history of seizures, no history of motion sickness.	-
Schneider 2007 [38]	*N* = 105 (Crossover design. Participants received the VR treatment during either their 1st CHT treatment (group A) or their 2nd CHT treatment (group B). During the alternate CHT treatment (CG) subjects received standard care)	range: 32–78	breast, lung, or colon cancer; no previous history of cancer, age ≥ 18, requires at least two matched cycles of IV CHT, reading and writing in English, no metastasis and primary brain disease, without a history of seizures, no history of motion sickness, consent to participate in the study	-
Schneider 2011 [39]	*N* = 137 (Crossover design; requiring two matched IV CHT treatments. Participants were randomly assigned to receive VR distraction intervention during the 1st or 2nd treatment and standard care with no distraction during the alternate treatment)	range: 27–78	breast, lung, or colon cancer; no previous history of cancer, age ≥ 18, requires at least two matched cycles of IV CHT, reading and writing in English, consent to participate in the study	metastasis and primary brain disease, history of motion sickness or seizures

Abbreviations: VR—virtual reality; IG—intervention group; CG—control group; SD—standard deviation; PNF—proprioceptive neuromuscular facilitation; MT—music therapy; CHT—chemotherapy; RT—radiotherapy; IV—intravenous.

**Table 3 ijerph-18-08768-t003:** Characteristics of aim, intervention, measures, and results of the included studies.

First Author and Year [No. Ref.]	Aim of the Study	Type of Intervention Using VR	Duration of Therapy	Tested Variables	Result
Atef 2020 [29]	Establishing and comparing PNF and VR therapy in the treatment of lymphoedema after mastectomy	non-immersive VR Nintendo Wii^®^ video game (exercises such as boxing, tennis, triceps extension)	2 sessions per wk. of 30 min, for 4 wks.	-Lymphedema: circumferential method: The excess arm volume (EAV) = VL − VH, where VL refers to the lymphoedematous limb’s volume, and VH refers to the healthy extremity’s volume. -Function: QuickDASH-9 scale	-In the VR group was a greater improvement in lymphedema and upper limb function than in the PNF group. -No statistically significant differences were found between EAVs and QuickDASH-9 scores between these groups. The results before and after therapy with the use of VR improved EAVs (*p* = 0.001) and QuickDASH-9 scores (*p* = 0.001)
Chirico 2019 [30]	Effectiveness and comparison of the effects of therapy with the use of VR and MT in alleviating the psychological stress associated with CHT in BC patients	Immersive and interactive VR. The VR equipment: head-mounted glasses (Vuzix Wrap 1200 VR) with a head motion tracking system. Relaxing images virtual surroundings created on the Second Life^®^ platform (Linden Lab), e.g., being on an island, walking through a forest, observing animals, climbing a mountain, and swimming in the sea.	20 min. during CHT treatment lasted 45–90 min.	-Anxiety levels: State Anxiety Inventory (SAI) for adults -Mood states: short version of Profile of Mood States (SV-POMS) -Cybersickness symptoms: The Virtual Reality Symptom Questionnaire (VRSQ)	VR therapy turned out to be more effective than MT. The anxiety decreased significantly in the CG group and was statistically insignificant in IG. Cybersickness symptoms occurred at a frequency of less than 20% (except for difficulty concentrating).
Feyzioğlu 2020 [31]	Evaluation of the impact of VR therapy with the use of Kinect equipment on the functions of the upper limb in women after BC surgery.	Kinect Sports I (boxing, darts, bowling,), Kinect Sports I (beach volleyball, table tennis), and Fruit Ninja.	35 min for one session during 6 wks.	-Pain intensity: VAS -Shoulder range of motion was measured in degrees using a digital goniometer -Arm strength was measured during maximal voluntary isometric muscle contraction with the J Tech Commender Muscle Tester handheld dynamometer -Handgrip strength was measured with the Saehan hydraulic hand dynamometer -The disability of the arm, shoulder, and hand (DASH) questionnaire was used to assess upper extremity functionality -Fear of movement as assessed with the Tampa Kinesiophobia Scale (TKS)	Kinect therapy was more effective in improving the assessed parameters than standard therapy (apart from the DASH results and the strength of the handgrip strength)
House 2016 [32]	BrightArm Duo therapy assessment in the context of coping with postoperative pain and disability after surgery treatment in BC patients with depression	The BrightArm Duo Rehabilitation System consisted of a low-friction robotic rehabilitation table, computerised forearm supports, a display, a laptop computer for the therapist station, a remote clinical server, and rehabilitation games. Nine games for manual motor training, cognitive and emotive training.	2 sessions (session lasted 20–50 min) per wk. for 8 wks	-Therapy session data consisted of supported arm reach baseline on the BrightArm Duo table (as measured by overhead digital cameras), power grasp strength baseline (as measured by a forearm support grasp sensor), HR and BP, number of active movements, and grasp repetitions for each arm during a session collected during play. -Pain: Numerical Rating Scale -The subjects rated their experience on a custom subjective questionnaire. The 10 questions were rated using a 5-point Likert scale; -Upper extremity function: the Fulg–Meyer Assessment, Upper Extremity Section, the Chedokee Arm and Hand Activity Inventory-9 (CAHAI-9) for bimanual tasks, and the Jebsen Hand Function Test (JHFT)24 for hand function. -Arm and hand range of motion: mechanical goniometers -Shoulder strength: wrist weights, -Grasp strength: a Jamar mechanical dynamometer and a pinch meter. -Independence in ADL involving the upper extremity: the upper extremity functional index 20 (UEFI-20). -Neuropsychological evaluations: the Beck Depression Inventory, Second Edition (BDI-II), the Neuropsychological Assessment Battery (NAB) Attention Module (orientation, digit span, and dots) and Executive Functioning Module (generation subtest), the Hopkins Verbal Learning Test, Revised (HVLT-R), the Brief Visuospatial Memory Test, Revised (BVMT-R), and the Trail Making Test (TMT) A and B.	There was a decrease in pain and severity of depression (*p* = 0.1; *p* = 0.04), an increase in 17 out of 18 motor indices, an improvement in 13 out of 15 indices of strength and function.
McGarvey 2009 [33]	using a computer-imaging program called help with adjustment to alopecia by image recovery (HAAIR) to provide educational support and reduce stress in women with post-chemotherapy hair loss	HAAIR system allows you to see yourself in VR with a bald head and of her head wearing a variety of different wigs and hairstyles	one session (60-90 min)	-Qualitative data were obtained verbally by open-ended questions asked during or following use of HAAIR (e.g., ‘Did you enjoy using the system?’, ‘Are you glad that you were able to try the system?’) -The quantitative measures: A Demographic Characteristics Questionnaire, The Brief Symptom Inventory (BSI-18), The Importance of Hair Questionnaire (IHQ), The Brief Cope, was used to assess the manner in which patients coped with having cancer.	Hair loss distress decreased in both the CG and IG groups at time after hair loss than at baseline with 3 months follow-up distress scores increasing in the SCG and decreasing in the IG. Those with avoidance coping reported more distress.
Mohammad 2018 [34]	Assessment of the effectiveness of immersive VR distraction technology in reducing pain and anxiety among female BC patients.	-Immersive VR -The IG chose from two scenarios on a CD-ROM, which included deep-sea diving ‘Ocean Rift,’ or sitting on the beach with the ‘Happy Place’ track. Then, the patients wore a head-mounted display with headphones. The VR exposure session was ended at the peak time of painkiller efficacy.	once	-Pain intensity: VAS -Anxiety: The State Anxiety Inventory (SAI)	-One session of the immersive VR plus morphine made a significant reduction in pain and anxiety self-reported scores, compared with morphine alone, in BC patients. -The independent-sample *t*-test showed a significant difference post intervention between the two groups’ pain scores. -The paired *t*-test showed a significant difference in the means of pain scores at the pre-and post-test in IG and the CG. -Regarding the anxiety testing, the independent sample *t*-test showed a significant difference post intervention between the two groups. The IG mean was lower than in the CG (*p* < 0.001).
Piejko 2020 [35]	Impact of medical resort treatment extended with modern feedback exercises using VR to improve postural control in BC survivors.	-Individual exercises were conducted using feedback based on VR and were aimed at improving motor coordination and body balance. Exercises on Alfa and Gamma stabilometric platforms and exercises of motor coordination of lower limbs with elastic resistance were used using the Telko device. Rehab software enabling feedback based on VR and collecting data on the type of tasks ordered to patients and the accuracy of their performance by patients.	3 weeks (6 d/wk, for 45 min./d)	-Static postural control was assessed in a Romberg test -Dynamic postural control was evaluated in the dynamic test, during which the patient’s task was to move the centre of gravity of the body in different directions in a targeted and controlled manner in accordance with the task displayed on the monitor screen.	-In the assessment of dynamic postural control, the length of the centre of foot pressure (COP) movement path before the treatment and after treatment was statistically significant (*p* = 0.0083) shortened. In the assessment of static postural control, no statistically significant differences were found between the length of the COP pathway before treatment compared to the condition before treatment (*p* > 0.05).
Schneider 2003 [36]	Answer the questions: -Is VR an effective distraction intervention for reducing CHT-related symptom distress levels in older women with BC? -Does VR have a lasting effect?	-The individual wears an 8-oz head-mounted device, which projects an image with the corresponding sounds. The sense of touch is involved through the use of a computer mouse that allows for the manipulation of the image. Participants chose from three CD-ROM-based scenarios; (Oceans Below^®^, A World of Art^®^, or Titanic: Adventure Out of Time^®^).	once during IV CHT	-Fatigue measures: Revised Piper Fatigue Scale (PFS) -Transitory anxiety states in adults: State-Anxiety Inventory for Adults (SAI) -Indicator of symptoms experienced by cancer patients Symptom Distress Scale (SDS)	Analysis using paired t-tests demonstrated a significant decrease in the SAI (*p* = 0.10) scores following CHT treatments when participants used VR. -No significant changes were found in SDS or PFS values. There was a consistent trend toward improved symptoms on all measures 48 h following completion of CHT. Evaluation of the intervention indicated that women t experienced no cybersickness, and 100% would use VR again.
Schneider 2004 [37]	To explore the use of VR as a distraction intervention to relieve symptom distress in women receiving CHT for BC.	For this study, a commercially available headset (Sony PC Glasstron PLM-S700) was used. Subjects chose from three scenarios on CD-ROM. Each scenario lasts several hours, and choices included deep-sea diving, walking through an art museum, or solving a mystery.	once during IV CHT	-Concerns of patients receiving CHT treatments: The Symptom Distress Scale (SDS) -Anxiety and fatigue: The State-Trait Anxiety Inventory (SAI) for Adults and Revised Piper Fatigue Scale (PFS) -Evaluation of VR intervention-open-ended questionnaire that was used to elicit subjects’ opinions about the intervention	Significant decreases in symptom distress and fatigue occurred immediately following CHT treatments when women used the VR intervention.
Schneider 2007 [38]	To explore VR as a distraction intervention to relieve symptom distress in adults receiving CHT for breast, colon, and lung cancer.	A commercially available headset (i-Glasses^®^ SVGA Head-Mounted Display, i-O Display Systems, Menlo Park, CA) was used. Participants chose from 4 possible CD-ROM–based VR scenarios: deep-sea diving (Oceans Below^®^, CounterTop Software, Renton, WA), walking through an art museum (A World of Art^®^, CounterTop Software), exploring ancient worlds (Timelapse^®^, Hammerhead Entertainment, Encinitas, CA), and solving a mystery (Titanic: Adventure Out of Time^®^, Hammerhead Entertainment).	once during IV CHT	-Validation of the distracting qualities of the intervention: Presence Questionnaire (PQ) and the Evaluation of Virtual Reality Intervention -Symptom distress: Adapted Symptom Distress Scale-2 (ASDS-2), State Anxiety Inventory for Adults (SAI), Revised Piper Fatigue Scale (PFS)	Individuals who received the VR during their first CHT had significantly less anxiety, compared with the control condition during the second CHT treatment.
Schneider 2011 [39]	Explore the influence of age, gender, state anxiety, fatigue, and diagnosis on time perception in cancer patients receiving IV CHT with a VR intervention within the cognitive model of time perception and predict the effects of these variables on the difference between the actual time elapsed while patients received CHT while immersed in a VR environment and their retrospective estimates of elapsed time.	The VR intervention delivered using commercially available HMDs, it was provided during the entire period of IV infusion, including delivery of pre-medications, antiemetics, and CHT agents. The researcher installed the HMD on the participant’s head as IV infusion started and removed the HMD as soon as the infusion process was completed. Patients selected an initial VR scenario from a menu of multiple options and were free to switch scenarios at any point during the treatment period.	once during IV CHT	-Anxiety: State-Trait Anxiety Inventory for Adults (STAI) -Fatigue: Revised Piper Fatigue Scale (PFS) -Actual time elapsed during CHT	In a forward regression model, three predictors (diagnosis, gender, and anxiety) explained a significant portion of the variability for altered time perception (*p* = 0.0008). Diagnosis was the strongest predictor; individuals with breast and colon cancer perceived time passed more quickly.

Abbreviations: VR—virtual reality; IG—intervention group; CG—control group; MT—music therapy; CHT—chemotherapy, MT—music therapy; IV—intravenous; ADL—activities of daily living; VAS—visual analogue scale.

## Data Availability

The datasets generated for this study are available on request to the corresponding author.

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
