# Peer review of "Virtual Reality as a Promising Tool Supporting Oncological Treatment in Breast Cancer"

_ijerph, 2021, doi:10.3390/ijerph18168768_

Round 1
Reviewer 1 Report
Dear Authors,
Thank you for the opportunity to review the paper entitled “Virtual Reality as a Promising Tool Supporting Oncological Treatment in Breast Cancer”. The presented paper addresses an important research area. I'm afraid that very limited keywords may have resulted in not including all the papers in this area. The article is methodologically correct, but I have some comments and suggestions that I think should be considered.
Minor
- The inclusion and exclusion criteria are repeated in the paper (lines 117-122 vs. 130-137)
- The paper does not identify the PICO(s) that appear in the register of the paper (reviewregistry1152)
- There is a mismatch in keywords between the manuscript and the registration
- When was performed the last search?
- The paper lacks a separate outcomes chapter where primary and secondary outcomes are listed
- I suggest changing the width of the tables to cover the whole width of the page - this is allowed in MDPI, it will make the tables easier to see, as they are narrow and long.
- A separate chapter on "excluded studies" is missing, describing why these manuscripts were rejected at the full-text stage.
- Individual paragraphs begin with different indentations (it looks like a single tab was used once, a double tab was used another time)
- I propose to standardize in Table 2, the age range of patients, once the average is given and once the range
- Line 229-231 the sentence "...users to sink themselves..." doesn't sound correct, I guess you mean the phenomenon of immersion, I suggest replacing it with: "...users to be immersed...".
- I don't understand why the authors start the discussion by pointing out the sedentary lifestyle? I don't see the connection to the purpose of the paper. The authors should start the discussion by describing the quality of the papers presented, indicate that some of the papers involved a small sample indicate the difference in age groups between the studies, etc. (Update: this was found at the very end of the paper, this is where the discussion should start)
- Lines 300-303 "The limitations found in the revised studies indicates that despite the in- creased use of VR technology in cancer rehabilitation, it is not possible to draw strong conclusions about the efficacy of VR-based rehabilitation for BC patients because of the overall lack of methodological quality and statistical power observed in the current literature”: The purpose of this study was not to evaluate the effectiveness of VR interventions in patients with breast cancer; as indicated in line 192-104 it was: “The aim of this systematic review was to conduct an overview of the clinical studies that used VR therapy in BC”. To my best knowledge, assessment of efficacy is done by conducting a meta-analysis, which this systematic review does not address - The sentence from the limitation needs to be changed.
- Lines 333-334: I have not noted any subjects of this study
Author Response
Thank you very much for your letter regarding our manuscript. Below, we have answered the Reviewers’ comments. Additionally, we attached the corrected version of the manuscript. We hope that in its amended form, the article is now suitable for publication.
Yours sincerely,
The authors
Ad:
- The inclusion and exclusion criteria are repeated in the paper (lines 117-122 vs. 130-137)
Answer: Thank you for your suggestions. You have right. The text has been changed according to reviewer’s comment (lines 117-122).
- The paper does not identify the PICO(s) that appear in the register of the paper (reviewregistry1152)
Answer: I am sorry, but we did not understand this comment. We did not use PICO.
- There is a mismatch in keywords between the manuscript and the registration
Answer: Thank you for your comment. We made a mistake while writing manuscript keywords. Sorry. We have corrected to match keywords.
- When was performed the last search?
Answer: The last database search was performed on 1 March 2021.
- The paper lacks a separate outcomes chapter where primary and secondary outcomes are listed
Answer; Thank you for suggesting the important information that should be included in this manuscript. We have added this information as separate chapter.
- I suggest changing the width of the tables to cover the whole width of the page - this is allowed in MDPI, it will make the tables easier to see, as they are narrow and long.
Answer: You have right. We have corrected the tables according to reviewer's guidelines.
- A separate chapter on "excluded studies" is missing, describing why these manuscripts were rejected at the full-text stage.
Answer: Articles were excluded because they did not meet the inclusion criteria. The reasons for the rejection of full- text articles have been supplemented in Figure 1.
- Individual paragraphs begin with different indentations (it looks like a single tab was used once, a double tab was used another time)
Answer:You are right, thank you. We have made a correction.
- I propose to standardize in Table 2, the age range of patients, once the average is given and once the range
Answer: We would like to thank you for your suggestions, but all data on the age of the subjects depends on how they are originally presented in the included articles. It was mean in some, range in others. Therefore, we presented so different informations from articles.
- Line 229-231 the sentence "...users to sink themselves..." doesn't sound correct, I guess you mean the phenomenon of immersion, I suggest replacing it with: "...users to be immersed...".
Answer: You are right, thank you. We have made a correction.
- I don't understand why the authors start the discussion by pointing out the sedentary lifestyle? I don't see the connection to the purpose of the paper. The authors should start the discussion by describing the quality of the papers presented, indicate that some of the papers involved a small sample indicate the difference in age groups between the studies, etc. (Update: this was found at the very end of the paper, this is where the discussion should start)
Answer: We absolutely agree with this statement. Thank you for the suggestions of start the Discussion part by describing the quality of the presented papers. We hope it is sufficient now.
- Lines 300-303 "The limitations found in the revised studies indicates that despite the in- creased use of VR technology in cancer rehabilitation, it is not possible to draw strong conclusions about the efficacy of VR-based rehabilitation for BC patients because of the overall lack of methodological quality and statistical power observed in the current literature”: The purpose of this study was not to evaluate the effectiveness of VR interventions in patients with breast cancer; as indicated in line 192-104 it was: “The aim of this systematic review was to conduct an overview of the clinical studies that used VR therapy in BC”. To my best knowledge, assessment of efficacy is done by conducting a meta-analysis, which this systematic review does not address - The sentence from the limitation needs to be changed.
Answer: The limitation paragraph in the Discussion has been heavily reedited according to your suggestions, thank you.
- Lines 333-334: I have not noted any subjects of this study
Answer: Thank you. We have changed this part of the conclusion. Maybe too far we came up with recommendations for using VR. We hope, we did not miss anything and explain well this part of the conclusion. Please check our correction.
Reviewer 2 Report
This paper is thought to be meaningful in that it summarizes the role of VR in the breast cancer area.
But overall, it looks distracting and the table needs to be corrected.
1. The roles of authors should be summarized later in the paper.
2. The following sentences overlap
'Seven studies were considered as "strong" [29–31,33–35,39] and four were considered a “weak" [32,36–38]'
The highest rated section was data collection, and the worst blinding and selection bias.
3. You need to adjust the size of the age column in Table2.
4. There are also some misspellings. Please confirm.
First autor -> First author
5. The full term for the abbreviation also needs to be added.
'VL' 'VH'
6. Table 3 is too distracting.
Author Response
Thank you very much for your letter regarding our manuscript. Below, we have answered the Reviewers’ comments. Additionally, we attached the corrected version of the manuscript. We hope that in its amended form, the article is now suitable for publication.
Yours sincerely,
The authors
Corrected according to the reviewer's guidelines:
- The roles of authors should be summarized later in the paper.
Answer: Thank you, we are made the change.
- The following sentences overlap
'Seven studies were considered as "strong" [29–31,33–35,39] and four were considered a “weak" [32,36–38]'
The highest rated section was data collection, and the worst blinding and selection bias.
Answer: You are right, thank you. We have changed it.
- You need to adjust the size of the age column in Table2.
Answer: You are right, thank you. We have made a correction.
- There are also some misspellings. Please confirm.
First autor -> First author
Answer: Sorry for those mistakes. We have corrected this information.
- The full term for the abbreviation also needs to be added.
'VL' 'VH'
Answer: This information has been corrected, thank you.
- Table 3 is too distracting.
Answer: Thank you for your suggestion. The width of Table 3 has been increased to make it easier to read.
Round 2
Reviewer 2 Report
The role of VR in the rehabilitation of breast cancer patients is well summarized. It will be helpful for doctors and therapists related to breast cancer.